# Adaptivity to Local Smoothness and Dimension in Kernel Regression

**Samory Kpotufe**
Toyota Technological Institute-Chicago*
samory@ttic.edu

**Vikas K Garg**
Toyota Technological Institute-Chicago
vkg@ttic.edu

## Abstract

We present the first result for kernel regression where the procedure adapts locally at a point $x$ to both the unknown local dimension of the metric space $\mathcal{X}$ and the unknown Hölder-continuity of the regression function at $x$. The result holds with high probability simultaneously at all points $x$ in a general metric space $\mathcal{X}$ of unknown structure.

## 1 Introduction

Contemporary statistical procedures are making inroads into a diverse range of applications in the natural sciences and engineering. However it is difficult to use those procedures "off-the-shelf" because they have to be properly tuned to the particular application. Without proper tuning their prediction performance can suffer greatly. This is true in nonparametric regression (e.g. tree-based, k-NN and kernel regression) where regression performance is particularly sensitive to how well the method is tuned to the unknown problem parameters.

In this work, we present an *adaptive* kernel regression procedure, i.e. a procedure which self-tunes, optimally, to the unknown parameters of the problem at hand.

We consider regression on a general metric space $\mathcal{X}$ of unknown *metric dimension*, where the output $Y$ is given as $f(x)$ + noise. We are interested in adaptivity at any input point $x \in \mathcal{X}$: the algorithm must self-tune to the unknown *local* parameters of the problem at $x$. The most important such parameters (see e.g. [1, 2]), are (1) the unknown smoothness of $f$, and (2) the unknown intrinsic dimension, both defined over a neighborhood of $x$. Existing results on adaptivity have typically treated these two problem parameters separately, resulting in methods that solve only part of the self-tuning problem.

In kernel regression, the main algorithmic parameter to tune is the bandwidth $h$ of the kernel. The problem of (local) bandwidth selection at a point $x \in \mathcal{X}$ has received considerable attention in both the theoretical and applied literature (see e.g. [3, 4, 5]). In this paper we present the first method which provably adapts to both the unknown local intrinsic dimension and the unknown Hölder-continuity of the regression function $f$ at any point $x$ in a metric space of unknown structure. The intrinsic dimension and Hölder-continuity are allowed to vary with $x$ in the space, and the algorithm must thus choose the bandwidth $h$ as a function of the query $x$, for all possible $x \in X$.

It is unclear how to extend global bandwidth selection methods such as cross-validation to the local bandwidth selection problem at $x$. The main difficulty is that of evaluating the regression error at $x$ since the ouput $Y$ at $x$ is unobserved. We do have the labeled training sample to guide us in selecting $h(x)$, and we will show an approach that guarantees a regression rate optimal in terms of the local problem complexity at $x$.

The result combines various insights from previous work on regression. In particular, to adapt to Hölder-continuity, we build on acclaimed results of Lepski et al. [6, 7, 8]. In particular some such *Lepski's adaptive methods* consist of monitoring the change in regression estimates $f_{n,h}(x)$ as the bandwidth $h$ is varied. The selected estimate has to meet some stability criteria. The stability criteria is designed to ensure that the selected $f_{n,h}(x)$ is sufficiently close to a target estimate $f_{n,\tilde{h}}(x)$ for a bandwidth $\tilde{h}$ known to yield an optimal regression rate. These methods however are generally instantiated for regression in $\mathbb{R}$, but extend to high-dimensional regression if the dimension of the input space $\mathcal{X}$ is known. In this work however the dimension of $\mathcal{X}$ is unknown, and in fact $\mathcal{X}$ is allowed to be a general metric space with significantly less regularity than usual Euclidean spaces.

To adapt to local dimension we build on recent insights of [9] where a $k$-NN procedure is shown to adapt locally to intrinsic dimension. The general idea for selecting $k = k(x)$ is to balance surrogates of the unknown bias and variance of the estimate. As a surrogate for the bias, nearest neighbor distances are used, assuming $f$ is globally Lipschitz. Since Lipschitz-continuity is a special case of Hölder-continuity, the work of [9] corresponds in the present context to knowing the smoothness of $f$ everywhere. In this work we do not assume knowledge of the smoothness of $f$, but simply that $f$ is locally Hölder-continuous with unknown Hölder parameters.

Suppose we knew the smoothness of $f$ at $x$, then we can derive an approach for selecting $h(x)$, similar to that of [9], by balancing the proper surrogates for the bias and variance of a kernel estimate. Let $\bar{h}$ be the hypothetical bandwidth so-obtained. Since we don't actually know the local smoothness of $f$, our approach, similar to Lepski's, is to monitor the change in estimates $f_{n,h}(x)$ as $h$ varies, and pick the estimate $f_{n,\hat{h}}(x)$ which is deemed close to the hypothetical estimate $f_{n,\bar{h}}(x)$ under some stability condition.

We prove nearly optimal local rates $\tilde{O}\left(\lambda^{2d/(2\alpha+d)} n^{-2\alpha/(2\alpha+d)}\right)$ in terms of the local dimension $d$ at any point $x$ and Hölder parameters $\lambda, \alpha$ depending also on $x$. Furthermore, the result holds with high probability, simultaneously at all $x \in \mathcal{X}$, for $n$ sufficiently large. Note that we cannot *union-bound* over all $x \in \mathcal{X}$, so the uniform result relies on proper conditioning on particular events in our variance bounds on estimates $f_{n,h}(\cdot)$.

We start with definitions and theoretical setup in Section 2. The procedure is given in Section 3, followed by a technical overview of the result in Section 4. The analysis follows in Section 5.

## 2 Setup and Notation

### 2.1 Distribution and sample

We assume the input $X$ belongs to a metric space $(\mathcal{X}, \rho)$ of bounded diameter $\Delta_{\mathcal{X}} \geq 1$. The output $Y$ belongs to a space $\mathcal{Y}$ of bounded diameter $\Delta_{\mathcal{Y}}$. We let $\mu$ denote the marginal measure on $\mathcal{X}$ and $\mu_n$ denote the corresponding empirical distribution on an i.i.d. sample of size $n$. We assume for simplicity that $\Delta_{\mathcal{X}}$ and $\Delta_{\mathcal{Y}}$ are known.

The algorithm runs on an i.i.d training sample $\{(X_i, Y_i)\}_{i=1}^n$ of size $n$. We use the notation $\mathbf{X} \doteq \{X_i\}_1^n$ and $\mathbf{Y} = \{Y_i\}_1^n$.

### Regression function

We assume the regression function $f(x) \doteq \mathbb{E}[Y|x]$ satisfies **local** Hölder assumptions: for every $x \in \mathcal{X}$ and $r > 0$, there exists $\lambda, \alpha > 0$ depending on $x$ and $r$, such that $f$ is $(\lambda, \alpha)$-Hölder **at** $x$ on $B(x,r)$:

$$\forall x' \in B(x,r) \quad |f(x) - f(x')| \leq \lambda \rho(x,x')^{\alpha}.$$

We note that the $\alpha$ parameter is usually assumed to be in the interval $(0,1]$ for global definitions of Hölder continuity, since a global $\alpha > 1$ implies that $f$ is constant (for differentiable $f$). Here however, the definition being given relative to $x$, we can simply assume $\alpha > 0$. For instance the function $f(x) = x^{\alpha}$ is clearly locally $\alpha$-Hölder at $x = 0$ with constant $\lambda = 1$ for any $\alpha > 0$. With higher $\alpha = \alpha(x)$, $f$ gets *flatter* locally at $x$, and regression gets easier.

**Notion of dimension**

We use the following notion of metric-dimension, also employed in [9]. This notion extends some global notions of metric dimension to local regions of space . Thus it allows for the intrinsic dimension of the data to vary over space. As argued in [9] (see also [10] for a more general theory) it often coincides with other natural measures of dimension such as manifold dimension.

**Definition 1.** *Fix $x \in \mathcal{X}$, and $r > 0$. Let $C \geq 1$ and $d \geq 1$. The marginal $\mu$ is $(C, d)$-**homogeneous on** $B(x, r)$ if we have $\mu(B(x, r')) \leq C\epsilon^{-d}\mu(B(x, \epsilon r'))$ for all $r' \leq r$ and $0 < \epsilon < 1$.*

In the above definition, $d$ will be viewed as the local dimension at $x$. We will require a general upper-bound $d_0$ on the local dimension $d(x)$ over any $x$ in the space. This is defined below and can be viewed as the worst-case intrinsic dimension over regions of space.

**Assumption 1.** *The marginal $\mu$ is $(C_0, d_0)$-**maximally-homogeneous** for some $C_0 \geq 1$ and $d_0 \geq 1$, i.e. the following holds for all $x \in \mathcal{X}$ and $r > 0$: suppose there exists $C \geq 1$ and $d \geq 1$ such that $\mu$ is $(C, d)$-homogeneous on $B(x, r)$, then $\mu$ is $(C_0, d_0)$-homogeneous on $B(x, r)$.*

Notice that if $\mu$ is $(C, d)$-homogeneous on some $B(x, r)$, then it is $(C_0, d_0)$-homogeneous on $B(x, r)$ for any $C_0 > C$ and $d_0 > d$. Thus, $C_0, d_0$ can be viewed as global upper-bounds on the local homogeneity constants. By the definition, it can be the case that $\mu$ is $(C_0, d_0)$-maximally-homogeneous without being $(C_0, d_0)$-homogeneous on the entire space $\mathcal{X}$.

The algorithm is assumed to know the upper-bound $d_0$. This is a minor assumption: in many situations where $\mathcal{X}$ is a subset of a Euclidean space $\mathbb{R}^D$, $D$ can be used in place of $d_0$; more generally, the global metric entropy (log of covering numbers) of $\mathcal{X}$ can be used in the place of $d_0$ (using known relations between the present notion of dimension and metric entropies [9, 10]). The metric entropy is relatively easy to estimate since it is a global quantity independent of any particular query $x$.

Finally we require that the local dimension is tight in small regions. This is captured by the following assumption.

**Assumption 2.** *There exists $r_\mu > 0, C' > 0$ such that if $\mu$ is $(C, d)$-homogeneous on some $B(x, r)$ where $r < r_\mu$, then for any $r' \leq r$, $\mu(B(x, r')) \leq C'r'^d$.*

This last assumption extends (to local regions of space) the common assumption that $\mu$ has an upper-bounded density (relative to Lebesgue). This is however more general in that $\mu$ is not required to have a density.

## 2.2 Kernel Regression

We consider a positive kernel $K$ on $[0, 1]$ highest at 0, decreasing on $[0, 1]$, and 0 outside $[0, 1]$. The kernel estimate is defined as follows: if $B(x, h) \cap \mathbf{X} \neq \emptyset$,

$$f_{n,h}(x) = \sum_i w_i(x)Y_i, \text{ where } w_i(x) = \frac{K(\rho(x, X_i)/h)}{\sum_j K(\rho(x, X_j)/h)}.$$

We set $w_i(x) = 1/n$, $\forall i \in [n]$ if $B(x, h) \cap \mathbf{X} = \emptyset$.

# 3 Procedure for Bandwidth Selection at $x$

**Definition 2.** *(Global cover size) Let $\epsilon > 0$. Let $\mathcal{N}_\rho(\epsilon)$ denote an upper-bound on the size of the smallest $\epsilon$-cover of $(\mathcal{X}, \rho)$.*

We assume the global quantity $\mathcal{N}_\rho(\epsilon)$ is known or pre-estimated. Recall that, as discussed in Section 2, $d_0$ can be picked to satisfy $\ln(\mathcal{N}_\rho(\epsilon)) = O(d_0 \log(\Delta_\mathcal{X}/\epsilon))$, in other words the procedure requires only knowledge of upper-bounds $\mathcal{N}_\rho(\epsilon)$ on global cover sizes.

The procedure is given as follows:

Fix $\epsilon = \frac{\Delta_\mathcal{X}}{n}$. For any $x \in \mathcal{X}$, the set of admissible bandwidths is given as

$$\hat{\mathcal{H}}_x = \left\{ h \geq 16\epsilon : \mu_n(B(x, h/32)) \geq \frac{32\ln(\mathcal{N}_\rho(\epsilon/2)/\delta)}{n} \right\} \bigcap \left\{ \frac{\Delta_\mathcal{X}}{2^i} \right\}_{i=0}^{\lceil \log(\Delta_\mathcal{X}/\epsilon) \rceil}.$$

Let $C_{n,\delta} \geq \frac{2K(0)}{K(1)} \left( 4 \ln \left( \mathcal{N}_\rho(\epsilon/2)/\delta \right) + 9C_0 4^{d_0} \right)$. For any $h \in \hat{\mathcal{H}}_x$, define

$$\hat{\sigma}_h = 2 \frac{\Delta_{\mathcal{Y}}^2 C_{n,\delta}}{n \cdot \mu_n(B(x, h/2))} \text{ and } D_h = \left[ f_{n,h}(x) - \sqrt{2\hat{\sigma}_h}, \, f_{n,h}(x) + \sqrt{2\hat{\sigma}_h} \right].$$

At every $x \in \mathcal{X}$ select the bandwidth:

$$\hat{h} = \max \left\{ h \in \hat{\mathcal{H}}_x : \bigcap_{h' \in \hat{\mathcal{H}}_x : h' < h} D_{h'} \neq \emptyset \right\}.$$

The main difference with Lepski's-type methods is in the parameter $\hat{\sigma}_h$. In Lepski's method, since $d$ is assumed known, a better surrogate depending on $d$ will be used.

## 4  Discussion of Results

We have the following main theorem.

**Theorem 1.** *Let $0 < \delta < 1/e$. Fix $\epsilon = \Delta_\mathcal{X}/n$. Let $C_{n,\delta} \geq \frac{2K(0)}{K(1)} \left( 9C_0 4^{d_0} + 4 \ln \left( \mathcal{N}_\rho(\epsilon/2)/\delta \right) \right)$.*

*Define $C_2 = \frac{4^{-d_0}}{6C_0}$. There exists $N$ such that, for $n > N$, the following holds with probability at least $1 - 2\delta$ over the choice of $(\mathbf{X}, \mathbf{Y})$, simultaneously for all $x \in \mathcal{X}$ and all $r$ satisfying*

$$r_\mu > r > r_n \triangleq 2 \left( \frac{2^{d_0} C_0^2 \Delta_\mathcal{X}^{d_0}}{C_2 \lambda^2} \right)^{1/(2\alpha + d_0)} \left( \frac{\Delta_{\mathcal{Y}}^2 C_{n,\delta}}{n} \right)^{1/(2\alpha + d_0)}.$$

*Let $x \in \mathcal{X}$, and suppose $f$ is $(\lambda, \alpha)$-Hölder at $x$ on $B(x, r)$. Suppose $\mu$ is $(C, d)$-homogeneous on $B(x, r)$. Let $C_r \doteq \frac{1}{C C_0 \Delta_\mathcal{X}^{d_0}} r^{d_0 - d}$. We have*

$$\left| f_{\hat{h}}(x) - f(x) \right|^2 \leq 96 C_0 2^{d_0} \cdot \lambda^{2d/(2\alpha + d)} \left( \frac{2^d \Delta_{\mathcal{Y}}^2 C_{n,\delta}}{C_2 C_r \lambda^2 n} \right)^{2\alpha/(2\alpha + d)}.$$

The result holds with high probability for all $x \in \mathcal{X}$, and for all $r_\mu > r > r_n$, where $r_n \xrightarrow{n \to \infty} 0$. Thus, as $n$ grows, the procedure is eventually adaptive to the Hölder parameters in any neighborhood of $x$. Note that the dimension $d$ is the same for all $r < r_\mu$ by definition of $r_\mu$. As previously discussed, the definition of $r_\mu$ corresponds to a requirement that the intrinsic dimension is *tight* in small enough regions. We believe this is a technical requirement due to our proof technique. We hope this requirement might be removed in a longer version of the paper.

Notice that $r$ is a factor of $n$ in the upper-bound. Since the result holds simultaneously for all $r_\mu > r > r_n$, the best tradeoff in terms of smoothness and size of $r$ is achieved. A similar tradeoff is observed in the result of [9].

As previously mentioned, the main idea behind the proof is to introduce hypothetical bandwidths $\bar{h}$ and and $\tilde{h}$ which balance respectively, $\hat{\sigma}_h$ and $\lambda^2 h^{2\alpha}$, and $O(\Delta_{\mathcal{Y}}^2/(nh^d))$ and $\lambda^2 h^{2\alpha}$ (see Figure 1). In the figure, $d$ and $\alpha$ are the unknown parameters in some neighborhood of point $x$.

The first part of the proof consists in showing that the variance of the estimate using a bandwidth $h$ is at most $\hat{\sigma}_h$. With high probability $\hat{\sigma}_h$ is bounded above by $O(\Delta_{\mathcal{Y}}^2/(nh^d))$. Thus by balancing $O(\Delta_{\mathcal{Y}}^2/(nh^d))$ and $\lambda^2 h^{2\alpha}$, using $\tilde{h}$ we would achieve a rate of $n^{-2\alpha/(2\alpha + d)}$. We then have to show that the error of $f_{n,\bar{h}}$ cannot be too far from that $f_{n,\tilde{h}}$.

Finally the error of $f_{n,\hat{h}}$, $\hat{h}$ being selected by the procedure, will be related to that of $f_{n,\bar{h}}$.

The argument is a bit more nuanced that just described above and in Figure 1: the respective curves $O(\Delta_{\mathcal{Y}}^2/(nh^d))$ and $\lambda^2 h^{2\alpha}$ are changing with $h$ since dimension and smoothness at $x$ depend on the size of the region considered. Special care has to be taken in the analysis to handle this technicality.

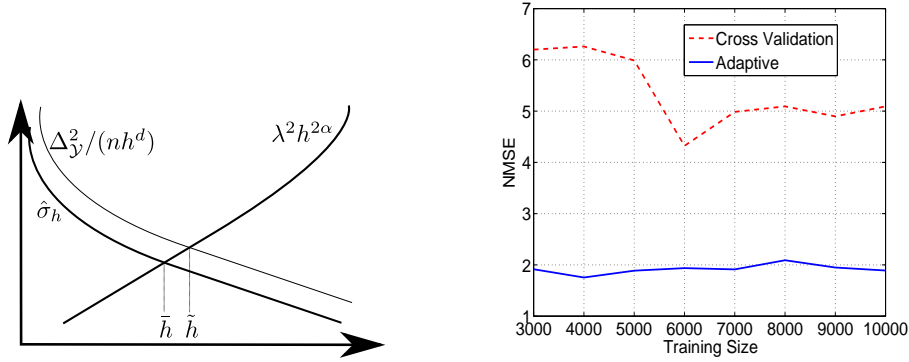

Figure 1: (Left) The proof argues over $\bar{h}$, $\tilde{h}$ which balance respectively, $\hat{\sigma}_h$ and $\lambda^2 h^{2\alpha}$, and $O(\Delta_\mathcal{Y}^2/(nh^d))$ and $\lambda^2 h^{2\alpha}$. The estimates under $\hat{h}$ selected by the procedure is shown to be close to that of $\bar{h}$, which in turn is shown to be close to that of $\tilde{h}$ which is of the right adaptive form.

(Right) Simulation results comparing the error of the proposed method to that of a global $h$ selected by cross-validation. The test size is 1000 for all experiments. $\mathcal{X} \subset \mathbb{R}^{70}$ has diameter 1, and is a collection of 3 disjoint flats (clusters) of dimension $d_1 = 2, d_2 = 5, d_3 = 10$, and equal mass $1/3$. For each $x$ from cluster $i$ we have the output $Y = (\sin \|x\|)^{k_i} + \mathcal{N}(0,1)$ where $k_1 = 0.8, k_2 = 0.6, k_3 = 0.4$. For the implementation of the proposed method, we set $\hat{\sigma}_h(x) = \hat{\text{var}}_Y/n\mu_n(B(x,h))$, where $\hat{\text{var}}_Y$ is the variance of $Y$ on the training sample. For both our method and cross-validation, we use a box-kernel, and we vary $h$ on an equidistant 100-knots grid on the interval from the smallest to largest interpoint distance on the training sample.

## 5   Analysis

We will make use of the the following bias-variance decomposition throughout the analysis. For any $x \in \mathcal{X}$ and bandwidth $h$, define the expected regression estimate

$$\widetilde{f}_{n,h}(x) \doteq \mathbb{E}_{\mathbf{Y}|\mathbf{X}} f_{n,h}(x) = \sum_i w_i f(X_i).$$

We have

$$|f_{n,h}(x) - f(x)|^2 \leq 2 \left|f_{n,h}(x) - \widetilde{f}_{n,h}(x)\right|^2 + 2 \left|\widetilde{f}_{n,h}(x) - f(x)\right|^2. \tag{1}$$

The bias term above is easily bounded in a standard way. This is stated in the Lemma below.

**Lemma 1** (Bias). *Let $x \in \mathcal{X}$, and suppose $f$ is $(\lambda, \alpha)$-Hölder at $x$ on $B(x,h)$. For any $h > 0$, we have $\left|\widetilde{f}_{n,h}(x) - f(x)\right|^2 \leq \lambda^2 h^{2\alpha}$.*

*Proof.* We have $\left|\widetilde{f}_{n,h}(x) - f(x)\right| \leq \sum_i w_i(x) |f(X_i) - f(x)| \leq \lambda h^\alpha$. $\qquad \square$

The rest of this section is dedicated to the analysis of the variance term of (1). We will need various supporting Lemmas relating the empirical mass of balls to their true mass. This is done in the next subsection. The variance results follow in the subsequent subsection.

### 5.1   Supporting Lemmas

We often argue over the following distributional counterpart to $\hat{\mathcal{H}}_x(\epsilon)$.

**Definition 3.** *Let $x \in \mathcal{X}$ and $\epsilon > 0$. Define*

$$\mathcal{H}_x(\epsilon) = \left\{ h \geq 8\epsilon : \mu(B(x, h/8)) \geq \frac{12 \ln(\mathcal{N}_\rho(\epsilon/2)/\delta)}{n} \right\} \bigcap \left\{ \frac{\Delta_\mathcal{X}}{2^i} \right\}_{i=0}^{\lceil \log(\Delta_\mathcal{X}/\epsilon) \rceil}.$$

**Lemma 2.** *Fix $\epsilon > 0$ and let $Z$ denote an $\epsilon/2$-cover of $\mathcal{X}$, and let $\mathcal{S}_\epsilon = \left\{ \frac{\Delta_\mathcal{X}}{2^i} \right\}_{i=0}^{\lceil \log(\Delta_\mathcal{X}/\epsilon) \rceil}$. Define $\gamma_n \doteq \frac{4\ln(\mathcal{N}_\rho(\epsilon/2)/\delta)}{n}$. With probability at least $1 - \delta$, for all $z \in Z$ and $h \in \mathcal{S}_\epsilon$ we have*

$$\mu_n(B(z,h)) \leq \mu(B(z,h)) + \sqrt{\gamma_n \cdot \mu(B(z,h))} + \gamma_n/3, \tag{2}$$

$$\mu(B(z,h)) \leq \mu_n(B(z,h)) + \sqrt{\gamma_n \cdot \mu_n(B(z,h))} + \gamma_n/3. \tag{3}$$

*Idea.* Apply Bernstein's inequality followed by a union bound on $Z$ and $\mathcal{S}_\epsilon$. $\qquad\square$

The following two lemmas result from the above Lemma 2.

**Lemma 3.** *Fix $\epsilon > 0$ and $0 < \delta < 1$. With probability at least $1 - \delta$, for all $x \in \mathcal{X}$ and $h \in \mathcal{H}_x(\epsilon)$, we have for $C_1 = 3C_0 4^{d_0}$ and $C_2 = \frac{4^{-d_0}}{6C_0}$,*

$$C_2\mu(B(x,h/2)) \leq \mu_n(B(x,h/2)) \leq C_1\mu(B(x,h/2)).$$

**Lemma 4.** *Let $0 < \delta < 1$, and $\epsilon > 0$. With probability at least $1-\delta$, for all $x \in \mathcal{X}$, $\hat{\mathcal{H}}_x(\epsilon) \subset \mathcal{H}_x(\epsilon)$.*

*Proof.* Again, let $Z$ be an $\epsilon/2$ cover and define $\mathcal{S}_\epsilon$ and $\gamma_n$ as in Lemma 2. Assume (2) in the statement of Lemma 2. Let $h > 16\epsilon$, we have for any $z \in Z$ and $x$ within $\epsilon/2$ of $z$,

$$\mu_n(B(x,h/32)) \leq \mu_n(B(z,h/16)) \leq 2\mu(B(z,h/16)) + 2\gamma_n \leq 2\mu(B(x,h/8)) + 2\gamma_n,$$

and we therefore have $\mu(B(x,h/8)) \geq \frac{1}{2}\mu_n(B(x,h/32)) - \gamma_n$. Pick $h \in \hat{\mathcal{H}}_x$ and conclude. $\qquad\square$

## 5.2 Bound on the variance

The following two results of Lemma 5 to 6 serve to bound the variance of the kernel estimate. These results are standard and included here for completion. The main result of this section is the variance bound of Lemma 7. This last lemma bounds the variance term of (1) with high probability simultaneously for all $x \in \mathcal{X}$ and for values of $h$ relevant to the algorithm.

**Lemma 5.** *For any $x \in \mathcal{X}$ and $h > 0$:*

$$\mathbb{E}_{\mathbf{Y}|\mathbf{X}} \left| f_{n,h}(x) - \widetilde{f}_{n,h}(x) \right|^2 \leq \sum_i w_i^2(x)\Delta_\mathcal{Y}^2.$$

**Lemma 6.** *Suppose that for some $x \in \mathcal{X}$ and $h > 0$, $\mu_n(B(x,h)) \neq 0$. We then have:*
$$\sum_i w_i^2(x) \leq \max_i w_i(x) \leq \frac{K(0)}{K(1) \cdot n\mu_n(B(x,h))}.$$

**Lemma 7** (Variance bound). *Let $0 < \delta < 1/2$ and $\epsilon > 0$. Define $C_{n,\delta} \doteq \frac{2K(0)}{K(1)} \left( 9C_0 4^{d_0} + 4\ln(\mathcal{N}_\rho(\epsilon/2)/\delta) \right)$, With probability at least $1 - 3\delta$ over the choice of $(\mathbf{X}, \mathbf{Y})$, for all $x \in \mathcal{X}$ and all $h \in \hat{\mathcal{H}}_x(\epsilon)$, $\left| f_{n,h}(x) - \widetilde{f}_{n,h}(x) \right|^2 \leq \frac{\Delta_\mathcal{Y}^2 C_{n,\delta}}{n\mu_n(B(x,h/2))}$.*

*Proof.* We prove the lemma statement for $h \in \mathcal{H}_x(\epsilon)$. The result then follows for $h \in \hat{\mathcal{H}}_x(\epsilon)$ with the same probability since, by Lemma 4, $\hat{\mathcal{H}}_x(\epsilon) \subset \mathcal{H}_x(\epsilon)$ under the same event of Lemma 2.

Consider any $\epsilon/2$-cover $Z$ of $\mathcal{X}$. Define $\gamma_n$ as in Lemma 2 and assume statement (3). Let $x \in \mathcal{X}$ and $z \in Z$ within distance $\epsilon/2$ of $x$. Let $h \in \mathcal{H}_x(\epsilon)$. We have

$$\mu(B(x,h/8)) \leq \mu(B(z,h/4)) \leq 2\mu_n(B(z,h/4)) + 2\gamma_n \leq 2\mu_n(B(x,h/2)) + 2\gamma_n,$$

and we therefore have $\mu_n(B(x,h/2)) \geq \frac{1}{2}\mu(B(x,h/8)) - \gamma_n \geq \frac{1}{2}\gamma_n$. Thus define $H_z$ denote the union of $\mathcal{H}_x(\epsilon)$ for $x \in B(z,\epsilon/2)$. With probability at least $1 - \delta$, for all $z \in Z$, and $x \in B(z,\epsilon/2)$,

and $h \in H_z$ the sets $B(z,h) \cap \mathbf{X}$, $B(x,h) \cap \mathbf{X}$ are all non empty since they all contain $B(x',h/2) \cap \mathbf{X}$ for some $x'$ such that $h \in \mathcal{H}_{x'}(\epsilon)$. The corresponding kernel estimates are therefore well defined. Assume w.l.o.g. that $Z$ is a minimal cover, i.e. all $B(z,\epsilon/2)$ contain some $x \in \mathcal{X}$.

We first condition on $\mathbf{X}$ fixed and argue over the randomness in $\mathbf{Y}$. For any $x \in \mathcal{X}$ and $h > 0$, let $Y_{x,h}$ denote the subset of $\mathbf{Y}$ corresponding to points from $\mathbf{X}$ falling in $B(x,h)$. We define $\phi(Y_{x,h}) \doteq \left| f_{n,h}(x) - \widetilde{f}_{n,h}(x) \right|$.

We note that changing any $Y_i$ value changes $\phi(Y_{z,h})$ by at most $\Delta_{\mathcal{Y}} w_i(z)$. Applying McDiarmid's inequality and taking a union bound over $z \in Z$ and $h \in H_z$, we get

$$\mathbb{P}(\exists z \in Z, \exists h \in S_\epsilon, \phi(Y_{z,h}) > \mathbb{E}\phi(Y_{z,h}) + t) \leq \mathcal{N}_\rho^2(\epsilon/2) \exp\left( -\frac{2t^2}{\Delta_{\mathcal{Y}}^2 \sum_i w_i^2(z)} \right).$$

We then have with probability at least $1 - 2\delta$, for all $z \in Z$ and $h \in H_z$,

$$\left| f_{n,h}(z) - \widetilde{f}_{n,h}(z) \right|^2 \leq 2\, \mathbb{E}_{Y|X}\left( \left| f_{n,h}(z) - \widetilde{f}_{n,h}(z) \right| \right)^2 + 2\ln\left( \frac{\mathcal{N}_\rho(\epsilon/2)}{\delta} \right) \Delta_{\mathcal{Y}}^2 \cdot \sum_i w_i^2(z)$$

$$\leq \left( 4\ln\left( \frac{\mathcal{N}_\rho(\epsilon/2)}{\delta} \right) \right) \cdot \frac{K(0)\Delta_{\mathcal{Y}}^2}{K(1) \cdot n\mu_n(B(z,h))}, \tag{4}$$

where we apply Lemma 5 and 6 for the last inequality.

Now fix any $z \in Z$, $h \in H_z$ and $x \in B(z,\epsilon/2)$. We have $|\phi(Y_{x,h}) - \phi(Y_{z,h})| \leq \max\{\phi(Y_{x,h}), \phi(Y_{z,h})\}$ since both quantities are positive. Thus $|\phi(Y_{x,h}) - \phi(Y_{z,h})|$ changes by at most $\max_{i,j}\{w_i(z), w_j(x)\} \cdot \Delta_{\mathcal{Y}}$ if we change any $Y_i$ value out of the contributing $Y$ values. By Lemma 6, $\max_{i,j}\{w_i(z), w_j(x)\} \leq \beta_{n,h}(x,z) \doteq \frac{K(0)}{nK(1)\min(\mu_n(B(x,h)), \mu_n(B(z,h)))}$. Thus define $\psi_h(x,z) \doteq \frac{1}{\beta_{n,h}(x,z)}|\phi(Y_{x,h}) - \phi(Y_{z,h}))|$ and $\psi_h(z) \doteq \sup_{x:\rho(x,z)\leq\epsilon/2} \psi_h(x,z)$. By what we just argued, changing any $Y_i$ makes $\psi_h(z)$ vary by at most $\Delta_{\mathcal{Y}}$. We can therefore apply McDiarmid's inequality to have that, with probability at least $1 - 3\delta$, for all $z \in Z$ and $h \in H_z$,

$$\psi_h(z) \leq \mathbb{E}_{\mathbf{Y}|\mathbf{X}}\psi_h(z) + \Delta_{\mathcal{Y}}\sqrt{\frac{2\ln(\mathcal{N}_\rho(\epsilon/2)/\delta)}{2n}}. \tag{5}$$

To bound the above expectation for any $z$ and $h \in H_z$, consider a sequence $\{x_i\}_1^\infty$, $x_i \in B(z,\epsilon/2)$ such that $\psi_h(x_i,z) \xrightarrow{i\to\infty} \psi_h(z)$. Fix any such $x_i$. Using Holder's inequality and invoking Lemma 5 and Lemma 6, we have

$$\mathbb{E}_{\mathbf{Y}|\mathbf{X}}\psi_h(x_i,z) = \frac{1}{\beta_{n,h}(x_i,z)}\mathbb{E}_{\mathbf{Y}|\mathbf{X}}|\phi(Y_{x_i,h}) - \phi(Y_{z,h})| \leq \frac{\sqrt{\mathbb{E}_{\mathbf{Y}|\mathbf{X}}(\phi(Y_{x_i,h}) - \phi(Y_{z,h})^2)}}{\beta_{n,h}(x_i,z)}$$

$$\leq \frac{\sqrt{2\mathbb{E}_{\mathbf{Y}|\mathbf{X}}\phi(Y_{x_i,h})^2 + 2\mathbb{E}_{\mathbf{Y}|\mathbf{X}}\phi(Y_{z,h})^2}}{\beta_{n,h}(x_i,z)} \leq \frac{\sqrt{4\Delta_{\mathcal{Y}}^2\beta_{n,h}(x_i,z)}}{\beta_{n,h}(x_i,z)}$$

$$= \frac{2\Delta_{\mathcal{Y}}}{\sqrt{\beta_{n,h}(x_i,z)}} \leq 2\Delta_{\mathcal{Y}}\sqrt{\frac{nK(1)\mu_n(B(z,h))}{K(0)}}.$$

Since $\psi_h(x_i,z)$ is bounded for all $x_i \in B(z,\epsilon)$, the Dominated Convergence Theorem yields

$$\mathbb{E}_{\mathbf{Y}|\mathbf{X}}\psi_h(z) = \lim_{i\to\infty}\mathbb{E}_{\mathbf{Y}|\mathbf{X}}\psi_h(x_i,z) \leq 2\Delta_{\mathcal{Y}}\sqrt{\frac{nK(1)\mu_n(B(z,h))}{K(0)}}.$$

Therefore, using (5), we have for any $z \in Z$, any $h \in H_z$, and any $x \in B(z,\epsilon/2)$ that, with probability at least $1 - 3\delta$

$$|\phi(Y_{x,h}) - \phi(Y_{z,h}))| \leq \Delta_{\mathcal{Y}}\beta_{n,h}(x,z)\left( 2\sqrt{\frac{nK(1)\mu_n(B(z,h))}{K(0)}} + \sqrt{\frac{2\ln(\mathcal{N}_\rho(\epsilon/2)/\delta)}{2n}} \right). \tag{6}$$

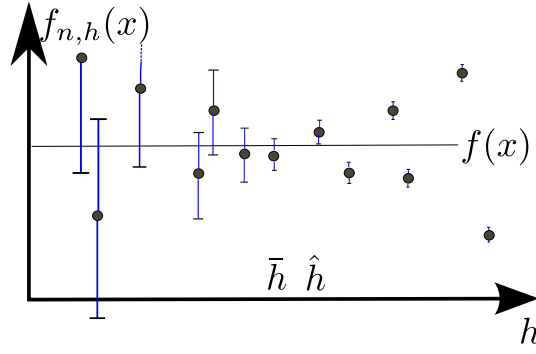

Figure 2: Illustration of the selection procedure. The intervals $D_h$ are shown containing $f(x)$. We will argue that $f_{n,\hat{h}}(x)$ cannot be too far from $f_{n,\bar{h}}(x)$.

Now notice that $\beta_{n,h}(x,z) \leq \dfrac{K(0)}{nK(1)\mu_n(B(x,h/2))}$, so by Lemma 3,

$$\mu_n(B(z,h)) \leq \mu_n(B(x,2h)) \leq C_1\mu(B(x,2h)) \leq C_1C_04^{d_0}\mu(B(x,h/2))$$
$$\leq C_2C_1C_04^{d_0}\mu_n(B(x,h/2)) \leq C_04^{d_0}\mu_n(B(x,h/2)).$$

Hence, (6) becomes $|\phi(Y_{x,h}) - \phi(Y_{z,h}))| \leq 3\Delta_{\mathcal{Y}}\sqrt{\frac{C04^{d_0}K(0)}{nK(1)\mu_n(B(x,h/2))}}$.

Combine with (4), using again the fact that $\mu_n(B(z,h)) \geq \mu_n(B(x,h/2))$ to obtain

$$\left|f_{n,h}(x) - \widetilde{f}_{n,h}(x)\right|^2 \leq 2\left|f_{n,h}(z) - \widetilde{f}_{n,h}(z)\right|^2 + 2\left|\phi(Y_{x,h}) - \phi(Y_{z,h}))\right|^2$$
$$\leq \frac{2\Delta_{\mathcal{Y}}^2}{n\mu_n(B(x,h/2))} \cdot \left(9C_04^{d_0} + 4\ln\left(\mathcal{N}_\rho(\epsilon/2)/\delta\right)\right).$$

$\square$

## 5.3 Adaptivity

The proof of Theorem 1 is given in the appendix. As previously discussed, the main part of the argument consists of relating the error of $f_{n,\bar{h}}(x)$ to that of $f_{n,\tilde{h}}(x)$ which is of the right form for $B(x,r)$ appropriately defined as in the theorem statement.

To relate the error of $f_{n,\hat{h}}(x)$ to that $f_{n,\bar{h}}(x)$, we employ a simple argument inspired by Lepski's adaptivity work. Notice that, by definition of $\hat{h}$ (see Figure 1 (Left)), for any $h \leq \bar{h}$ $\hat{\sigma}_h \geq \lambda^2 h^{2\alpha}$. Therefore by Lemma 1 and 7 that, for any $h < \bar{h}$, $\|f_{n,h} - f\|^2 \leq 2\hat{\sigma}_h$ so the intervals $D_h$ must all contain $f(x)$ and therefore must intersect. By the same argument $\hat{h} \geq \bar{h}$ and $D_{\hat{h}}$ and $D_{\bar{h}}$ must intersect. Now since $\hat{\sigma}_h$ is decreasing, we can infer that $f_{n,\hat{h}}(x)$ cannot be too far from $f_{n,\bar{h}}(x)$, so their errors must be similar. This is illustrated in Figure 2.

## Footnotes

*Other affiliation: Max Planck Institute for Intelligent Systems, Germany

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
