[Reviews · NeurIPS 2013]

Submitted by Assigned_Reviewer_6

The paper describes a method for local bandwidth selection in kernel regression models, which ensures adaptivity to local smoothness and dimension.

Quality: The paper presents a useful result for adaptivity in kernel regression. The work is set out well. I think that it would be useful to have some more discussion of the bandwidth selection procedure. How does \hat\mathcal{H}_x arise? And how easy is it to calculate \hat{h} in practice?

Clarity: The paper is clearly and concisely written.

Originality: The paper develops an original approach to the problem of local bandwidth selection.

Significance: I'm not an expert in this field so it is hard to come a decision about the paper's significance but the result seems very useful to me.
Summary: A well-written and clear paper with good potential for application.

Submitted by Assigned_Reviewer_7

This paper theoretically analyzes adptivity of kernel regression to
local smoothness and dimension. The authors consider a simple kernel
regressor (in Section 2.2), and developed a procedure for bandwidth
selection in Section 3. However, the rest of the paper is simply
devoted to proving Theorem 1; even Section 4 does not really discuss
the results, but describes a flow of the proof. Rather than detailed
proofs (which may be all included in the supplementary material), I
would like to see more discussion on outcomes of the results, possibly
with numerical examples.

Likewise, no conclusion is given, so the message of the paper is not
clear in the end.

***

In the authors' feedback, they promised to extend the discussion of the results and algorithms
to better introduce the topic of adaptivity to a general public.
I believe this will strengthen the paper significantly.
Summary: This paper is too mathematical and not suitable to NIPS. I suggest
the authors to submit this to a mathematical/statistics journal.

Submitted by Assigned_Reviewer_9

This paper considers the noisy regression setting with n data (X_i,Y_i) such that \E(Y | X=x) =f(x). The underlying function f is locally Holder smooth with associated coefficient \alpha_x. The distribution \mu underlying X is locally d_x dimensional.
The paper proposes a method to adapt locally the bandwidth in kernel regression to the unknown \alpha_x and d_x. The authors propose an adaptive estimate \hat f of f that is such that, if n large enough, with probability 1 - \delta, for any x
|\hat f(x) - f| < C (\log(n/\delta)/n)^{\alpha_x/(2\alpha_x + d)}.
They assume however that:
(i) An upper bound on \sup_x d_x is available
(ii) For each x, the dimension d_x is "attained" provided that we consider a small enough neighbourhood

This paper is a combination of results in [4] and [5] (see bibliography in the paper), i.e. local Lepski's method, and [9], i.e. adaptation to the intrinsic dimension: the results are interesting but not surprising in view of these works. I did not read the proofs in full details, but what I checked is correct and the results are plausible.
I have some questions:
- Results concerning adaptive (to the smoothness) methods such as Lepski's method are in general proved uniformly on a class of function (The alpha Holder functions in your case). Could you do the same?
- What about confidence intervals for your adaptive estimate?
- You assume (ii) for the intrinsic dimension. Do you really need this? It is not necessary in Lepski's method to assume that the smoothness \alpha is available: adaptive methods will tend to over-smooth, but this is not a problem. Could you have something similar with the intrinsic dimension?

To me, the main problem with this paper is the writing, which is not precise and not intuitive. Here is an un-exhaustive list of the issues I spotted:
-p1l43: the minimax optimal oracle rate is indeed n^{-2\alpha/(2\alpha+d)}, but you should definitely mention the minimax optimal adaptive rate, which is more relevant to your problem, i.e.
\min_{\hat f} \max_{\alpha} \max_{f \alpha Holder} \E \|\hat f - f\|_{\infty} (n/\log(n))^{2\alpha/(2\alpha+d)} > C
where C is a constant. In other words, the minimax optimal oracle rate is n^{-2\alpha/(2\alpha+d)}, but the minimax optimal rate is (n/\log(n))^{2\alpha/(2\alpha+d)} (inevitable additional (\log(n))^{2\alpha/(2\alpha+d)} to pay for the non-knowledge of \alpha). Also, it is the rate you obtain in your result.
-p2l64 and 76: Are \rho and \|.\| the same connected/ So \rho does actually define a norm?
-p2l90: What is B(x,r)? I assumed it is the ball of centre x and radius r but you should state it.
-p2l98: What is w.l.o.g.? Without loss of generality?
-p3l150: the constant N is very important. Could you give it explicitly?
-p3l161 (main equation): You have C_{n,\delta} = C\log(n/\delta): there is a dependency in \log(n)^{2\alpha/(2\alpha+d)} in your bound. Please make it explicit. This is anyway matching the lower (non-oracle) bound.
-p5l228: Please provide a proof for Lemma 2: shouldn't an additional log(1/(\epsilon\delta)) appear in the bound due to union bound on S_{\epsilon}?
-p6l286: citation for McDiarmid inequ please
-p7l153: The proof of Thm 1 is to me correct but redacted in a non precise way. Instead of saying "Assume that the statement of Lemma ..." is verified, it would be more clear to precise exactly on which event the Theorem holds.
-p8l395: Also the proof seems un-necessary complicated: you could maybe skip the definition of \bar h and prove that: (1) if c\tilde h < \hat h, then |\hat f_{n, \hat h}(x) - f(x)|< \hat \sigma_{\tilde h} < C \tilde R(\tilde h) and (2) the probability that c\tilde h > \hat h is very small since it is not likely that |\hat f_{n, c\tilde h} - \hat f_{n, \hat h}| > \hat \sigma_{\hat h} for such a \hat h.

Unify the notation log and ln. Is your log a log in base 2?

And also it would be great if you could extend the discussion on your results, and the intuitive explanations. The paper is to my mind quite hard to read, because very technical (sometimes without necessity). Also you should add some references on Lepski's method: the upper and lower bounds for this method were not provided in [4], [5] but before.
Summary: The results in this paper are interesting but the writing should really be improved. Provided that this is done, I would recommend acceptance of this paper.

Submitted by Assigned_Reviewer_10

This paper extends Lepski's method
to develop a nonparametric regression method
that adapts locally to dimension and smoothness.

Positives:

-Interesting problem
-Impressive theoretical analysis

Negatives:

-The assumption that the covering number is exactly known
(or not hugely over-estimated) seems very unrealistic

-There are no examples so it hard to judge whether this
method actually works in practice
Summary: Interesting idea but the paper seems incomplete.
Author Feedback

Author rebuttal: We thank the reviewers for the clear and pointed comments. The reviewers suggest moving more of the technical details into an appendix and extending the discussion of the results and algorithms. This is a good suggestion and we will follow it, especially to better introduce the topic of adaptivity to
a general public, stress the optimality of our bounds, and stress the topic's relevance to pattern recognition in general and machine learning in particular.
The topic of tuning is an important aspect of learning and especially nonparametric learning, and adaptive methods (self-tuning) are ever more relevant given the wider modern applicability of basic nonparametric methods such as kernel
regression.

Please find short answers to some specific concerns below.

++++++++++++++++++++++++++++++++++++++++++++++++++++++++++++++++++++++++++++++++++++++
Reviewer 10:
Our assumptions are actually quite general and cover for instance all the situations covered by [9], including data on manifold, sparse datasets, and mixtures of distributions on manifolds in general.

The assumption of bounded global covering number is a weak one in practice: we just need to know the global dimension d_0. In most practical settings this corresponds to the number of features. In more general metrics, the global
dimension can be estimated and many results are devoted to this global estimation (see [10] for a nice overview).

+++++++++++++++++++++++++++++++++++++++++++++++++++++++++++++++++++++++++++++++++++++++
Reviewer 6 and 7:
We will allocate more room to discussing the procedure and intuition. We point out to reviewer 7 that a lot of the intuition behind the procedure is mathematical in
nature, hence the nature of Section 4.

++++++++++++++++++++++++++++++++++++++++++++++++++++++++++++++++++++++++++++++++++++++++
Reviewer 9:
- It should be easy to give the bounds uniformly over a properly defined functional class (likely depending on d_0). Our bounds are given right now in a PAC style, uniformly for all x, f fixed, but the procedure only depends on d_0 being specified.

- The bounds of Theorem 1 being in terms of the unknown \alpha_x and d_x, confidence bands are not automatic.
However, in light of the selection procedure for h and key steps in the analysis, it might be possible to argue
that a (adaptive) confidence band might be of the form O(\hat{\sigma}_{\hat{h}}). This is a very interesting
question.

- We were not able to get around needing n large enough (at least for a general metric \X) for local regions
to have enough samples for d_x to be effectively 'attained'. However, as is likely clear to the reviewer, d_x is unknown to the algorithm. Work on adaptivity to (global) manifold dimension such as [Bikel and Li] are stated in a similar way as ours.

We will add more discussion and clarify various statements. For instance the metric space \X need not be normed,
and this will be made clear (we used \|.\| and \rho interchangeably which is confusing and bad style).
The union bound in Lemma 2 is over a set of total size at most (N_\rho(\epsilon/2))^2 and is reflected in the bound.

We are not sure the proof of Theorem 1 is easily shortened. This is actually the 3rd and simplest version we wrote.
However clarity can be improved by clearly restating important events from other lemmas.